# Genomic Analysis of ESBL-Producing *E. coli* in Wildlife from North-Eastern Germany

**DOI:** 10.3390/antibiotics11020123

**Published:** 2022-01-18

**Authors:** Timo Homeier-Bachmann, Anne K. Schütz, Sylvia Dreyer, Julien Glanz, Katharina Schaufler, Franz J. Conraths

**Affiliations:** 1Friedrich-Loeffler-Institut, Institute of Epidemiology, 17493 Greifswald–Insel Riems, Germany; Anne.Schuetz@fli.de (A.K.S.); Julien.Glanz@lazbw.bwl.de (J.G.); franz.conraths@fli.de (F.J.C.); 2Friedrich-Loeffler-Institut, Institute of International Animal Health/One Health, 17493 Greifswald–Insel Riems, Germany; sylvia.dreyer@fli.de; 3Wildlife Research Unit, Agricultural Centre Baden-Württemberg, 88326 Aulendorf, Germany; 4Institute of Pharmacy, University of Greifswald, 17489 Greifswald, Germany; katharina.schaufler@uni-greifswald.de; 5Institute of Infection Medicine, Christian-Albrecht University and University Medical Center Schleswig-Holstein, 24118 Kiel, Germany

**Keywords:** ESBL-*E. coli*, wild boar, wild ruminant, wildlife, AMR, resistome

## Abstract

Antimicrobial resistance (AMR) is a serious global health threat and extended-spectrum beta-lactamase (ESBL)-producing *Enterobacterales* are a major contributor. This study aimed to gain a deeper insight into the AMR burden of wild animals. In total, 1595 fecal samples were collected by two systematic searches in Mecklenburg-Western Pomerania, north-east Germany. Samples were screened for ESBL-carrying *Escherichia (E.) coli* and isolates found were further analyzed using antimicrobial susceptibility testing and whole-genome sequencing. We found an estimated prevalence of 1.2% ESBL-producing *E. coli* in wild boar and 1.1% in wild ruminants. CTX-M-1 was the most abundant CTX-M type. We also examined fecal samples from wild boar and wild ruminants using shotgun metagenomics to gain insight into the resistome in wild animals. The latter revealed significantly lower normalized counts for AMR genes in wildlife samples compared to farm animals. The AMR gene levels were lower in wild ruminants than in wild boar. In conclusion, our study revealed a low prevalence of ESBL-producing *E. coli* and a low overall AMR gene burden in wild boar and wild ruminants, probably due to the secluded location of the search area.

## 1. Introduction

The treatment of bacterial infections depends heavily on the availability of effective antimicrobial agents. Worldwide, the emergence and spread of antimicrobial resistance (AMR) is considered a serious public health threat [1]. In addition, the consumption of antimicrobials can result in alterations in the gut microbiota of humans and animals, leading to the selection of antimicrobial-resistant bacteria (AMRB). With urine and feces, AMRB and partially metabolized antimicrobials can enter wastewater and may subsequently be released via sewage systems into surface waters and cropland [2,3]. 

Antibiotic resistance is a natural phenomenon [4]; however, one of the main factors selecting for the massive emergence of AMR is misuse and overuse of antibiotics [5]. The AMR problem is not limited to humans, since part of the resistance burden in humans is influenced by the use of antimicrobials in livestock [6]. Between 2011 and 2018, an overall decrease in sales of antibiotics in veterinary medicine was observed in Europe. However, the broad use of antimicrobials—for example, in pig farming—is still ongoing [7]. Tetracyclines, amoxicillin, macrolides and colistin are most frequently used in pork production [8]. Antimicrobials are used in livestock production to treat individual animals. Besides, they are also administered for the treatment of whole groups, herds or flock, if clinical disease occurs in a small number of animals (metaphylactic approach) [8,9]. 

According to the WHO’s “Global Priority List of Antibiotic-Resistant Bacteria for Research, Discovery and Development of New Antibiotics”, extended-spectrum β-lactamase (ESBL)-producing *Enterobacterales* (ESBL-PE) are among the most critical antibiotic-resistant bacteria [10]. The enzymes produced by ESBL-carrying bacteria were first described in 1983. They hydrolyze beta-lactam antibiotics up to third- and fourth-generation cephalosporins such as cefotaxime [11]. There are several ESBL variants. The first were TEM (termed after the patient [Temoniera] from which the plasmid was isolated) and SHV (named after sulfhydryl-variable), which have since declined, while bacteria producing CTX-M-type β-lactamases (named after cefotaxime) are currently predominant. ESBL are mostly plasmid-mediated [12] and ESBL-PE exhibit a high zoonotic potential [10].

Of particular concern is the emergence of Gram-negative bacteria that are resistant to three of the most important antimicrobial classes (penicillins, cephalosporins, and quinolones [3MRGN = multidrug-resistant gram negatives]). 3MRGN occur in human healthcare settings [13], as well as in livestock, e.g., in poultry [14,15] and pigs [16]. ESBL-PE are often found to carry mobilizable colistin resistance genes (mcr) [17,18]. Although the agent colistin exhibits nephrotoxic and neurotoxic properties, it had to be re-introduced for the treatment of multidrug-resistant *Acinetobacter baumannii*, *Pseudomonas aeruginosa* and carbapenemase-producing *Enterobacteriaceae* infections in humans [19]. In recognition of the importance of colistin, the WHO has included the agent into the group of “highest priority critically important antimicrobials” in human medicine [20]. 

In general, wild animals do not come into direct contact with antimicrobials. Nevertheless, they may be exposed to AMRB or antimicrobial residues by foraging and drinking in environments contaminated with anthropogenic sources [3,16,21]. Several studies demonstrated that wildlife populations living in close proximity to humans and agricultural areas exhibited a higher prevalence of AMRB than those living in more pristine areas [6]. The large home range, omnivorous diet, and high tolerance of wild boar to human proximity may result in exposure to AMRB as well as antimicrobial residues. Consequently, this wildlife species has been suggested as a potential sentinel species for AMR in wildlife [6]. In addition, several studies have previously reported the occurrence of identical clones of ESBL-producing *E. coli* among humans, livestock and wildlife, reinforcing the role of these bacteria in transmission events, not only in veterinary settings, but also in public health, and vice versa [22,23]. However, there are still many open questions regarding the exact role that wildlife plays, for example, in the context of the prevalence of AMRB in wild animals and associated infections in humans [24]. Published studies primarily involve wild boar, and the most commonly studied bacterial species is *E. coli* [6]. 

Despite the increasing attention that ESBL-carrying *E. coli* in wild boar have recently received, only a few whole-genome sequencing (WGS) data sets are available on these resistant bacteria. With the present study, we aimed to fill this knowledge gap.

Of course, ESBL-bearing *E. coli* represent only one example of antibiotic-resistant bacteria. In addition to the aforementioned studies, which are based on analyses of individual isolates-, resistance genes can be detected using sequence-based methods. Metagenome analyses offer the possibility to characterize the bacterial resistome and provide insight into the abundance and diversity of antimicrobial resistance genes (AMRG). For urban wastewater, there is a strong correlation with socioeconomic, health, and environmental factors [25,26]. Recently, Liu et al. investigated the development of the resistome of dairy cows in the early life stage using metagenomic sequencing. The results suggest that modifications in the resistome may be possible via early-life dietary interventions to reduce overall AMRG [27]. A recent study, the first to present metagenomic data from pooled fecal samples of wild animals (red fox, wild boar, and rodent), demonstrated that AMRG levels are higher in livestock than in wild animals. Nevertheless, the results also show the potential of wild animals for AMR spread [28].

Therefore and in accordance with the list of “critically important antimicrobials for human medicine” published by the WHO [29] as well as the OIE List of Antimicrobial Agents of Veterinary Importance [30], this study aimed at analyzing ESBL-producing *E. coli* isolates and the resistomes collected from wildlife in a rather limitedly investigated area in the northeast of Germany. We particularly addressed (i) the geno- and phenotypic resistance profiles of ESBL-producing *E. coli*, (ii) the phylogenetic (sequence types) and virulence characteristics of these bacteria to place them into a broader context and assess, whether they belong to clonal lineages, associated with a high-risk at the international level, and (iii) the microbiome of selected fecal samples to reveal their respective resistome. We hypothesized that the resistance burden in wildlife is generally lower in wildlife than in farm animals in the same area. 

## 2. Results

### 2.1. Bacterial Isolation

Fecal samples from wild boar and wild ruminants were collected during two periods in February 2020 and 2021. The fresh samples were subsequently analyzed in the laboratory for the presence of ESBL-producing *E. coli*. 

In 2020, a total of 16 putative ESBL-*E. coli* were isolated from 820 fecal samples. Of these, 13 isolates originated from feces of wild boar (*n* = 693) and three from wild ruminants (*n* = 127). In 2021, only three isolates were obtained, and these were all isolated from wild boar fecal samples (*n* = 623). In 152 samples of wild ruminant feces, we did not find any ESBL-producing *E. coli*.

Thus, an estimated prevalence of 1.2% (confidence interval (CI) 0.7–2.0%; 16 of 1316) of ESBL-producing *E. coli* in wild boar and 1.1% (CI 0.2–3.1%, 3 of 279) in wild ruminants was calculated. For the individual collection periods, this results in 1.9% (13 of 693) positive fecal samples for wild boar in 2020 and 0.5% (3 of 623) in 2021. For wild ruminants, 2.4% (3 of 127) of samples were positive in 2020 and none (0 of 152) in 2021.

### 2.2. Antimicrobial Susceptibility Testing (AST)

All 19 isolates were resistant to the beta-lactam antibiotics amoxicillin/clavulanic acid, ampicillin, cefotaxime, ceftazidime, and cefepim. Six isolates were resistant to tetracycline (32%). In addition, two of the three isolates from 2021 (numbers 1031 and 1114) exhibited resistance to gentamicin and tobramycin, whereas the third isolate from 2021 (number 1115) was resistant to ciprofloxacin and trimethoprim/sulfamethoxazole. All isolates were sensitive to imipenem, meropenem, colistin, and amikacin. Thus, six of the 19 isolates phenotypically fulfilled the definition of multidrug-resistant (resistant to at least three classes of antibiotics). Details are given in Table 1. 

### 2.3. Whole-Genome Sequencing and Analysis

By analyzing WGS data obtained for the 19 suspected ESBL isolates, we identified three different phylotypes (A, B1, and E), all of which are mainly associated with commensal *E. coli* [31]. MLST (multi-locus sequence typing) analysis revealed the occurrence of five different sequence types (ST). ST216 was most common (*n* = 9), followed by ST398 (*n* = 4). ST543 appeared three times and ST2768 two times, ST6448 was found only once. Interestingly, ST398 appeared in both sampling periods, but only in wild boar samples. In contrast, ST216 and ST543 were identified in different species (wild boar and wild ruminants), but only during the first sampling round. 

Bacterial AMR evolves constantly and horizontal gene transfer through plasmids plays an important role in the spread of AMR [32]. We analyzed the carriage of plasmid replicons in each isolate [33]. Incompatibility (Inc) N plasmids were detected in all 2020 isolates, but these did not occur in isolates one year later. Also, exclusively in 2020, IncFIA-type plasmids were found in nine of 13 isolates (69%). The 2020 isolates that carried an IncFIA plasmid in addition contained IncFIB plasmids. The latter also occurred in two of the three 2021 isolates. These two 2021 isolates were also positive for IncFII and ColRNAI. Interestingly, sequencing of isolate 1115 from 2021 revealed no evidence of plasmid replicons. 

Beta-lactamases of the CTX-M type were present in all isolates (CTX-M1 and 55), other class A enzymes were genotypically absent as were plasmid-encoded beta-lactamases of classes B-D. By contrast, the chromosomally encoded beta-lactamases (AmpC and AmpH) were present in all isolates. Genes conferring resistance to aminoglycoside (e.g., *strA*, *strB*) antibiotics were detected in three isolates (1031, 1114 and 1115) by sequence analysis. These three isolates were obtained from wild boar samples collected in 2021. However, phenotypically, resistance to tobramycin and gentamicin was present only in isolates 1031 and 1114. Resistance genes conferring tetracycline resistance (*tetA*, *tetR*) were present in three isolates (1031, 1114, and 1115), while phenotypic resistance was found in three additional isolates (258, 259, and 261). Isolate 1115 was resistant to trimethoprim/sulfonamide both phenotypically and genotypically (*sul2*, *dfrA17*), whereas isolates 1031 and 1114 carried resistance genes to sulfonamides only (*sul1*, *sul2*) (sulfonamides, however, were not tested as single compounds in the AST). Resistance to phenicol (*floR*, *cmlA*) antibiotics was detected genotypically in the three isolates (257, 1031, and 1115). 

To gain insight into the virulence of the isolates, we searched for virulence genes associated with pathogenic bacteria. The main focus was on iron acquisition and resistance to heavy metals. We found a number of genes that play a role in iron uptake, such as enterobactin (*ent*), and heme receptors (*chu*). Furthermore, we found genes that confer resistance to heavy metals (e.g., *ars*, *znt*). In addition, we detected a few genes typical for extra-intestinal pathogenic *E. coli* (ExPEC). Details can be found in Figure 1.

### 2.4. Metagenomic Sequencing and Analysis of Antimicrobial Resistance Genes

To gain a comprehensive insight into the resistome of wild animals, we examined fecal samples from two wild boar and two wild ruminants using shotgun metagenomics. The generated metagenome data sets yielded between 2.23 and 3.45 billion base pairs, or 8.9 to 13.8 million reads per sample. Using the AMR++ pipeline with the MEGARes database, we detected 189 and 254 different AMRGs covering 15 and 17 AMR classes in the wild boar datasets, respectively. Resistance against bacitracin and fusidic acid occurred only in one of the samples. Comparable values were found in wild ruminants, i.e., 154 and 232 different AMRGs covering 16 and 17 AMR classes, respectively. Genes conferring resistance to sulfonamides were only detected in one sample. In addition, aminoglycoside resistance genes were slightly more common in the roe deer samples and, in contrast, resistance genes to tetracyclines were more common in wild boar (Table 2), the two species showed similar distributions of resistance genes (Table 2). 

Based on the 16S rRNA gene content, we found an overall abundance of 0.042 and 0.049 AMRG copies per 16S rRNA gene in wild boar samples. For the wild ruminant samples, we found slightly lower values of 0.028 and 0.033 AMRG copies per 16SrRNA gene, respectively (Table 3). In a control study conducted with a fecal sample of a calf from a farm in the same region, we found a normalized total AMRG abundance of 2.22.

### 2.5. Microbiome Analysis for Bacterial Species Diversity

To obtain a complete overview of the entire gut microhabitat of wild boar and wild ruminants, we performed a microbiome analysis based on the metagenome datasets. For this purpose, we used the CCMetagen pipeline [34]. For the wild boar metagenome datasets, we identified 14,817 and 12,073 reads matching with bacteria-derived sequences in the reference database. For the wild ruminant datasets, we allocated 4426 and 6412 reads in the reference database to bacterial origin, respectively. The most common bacterial families were *Bacteroidaceae*, *Oscillospiraceae* and *Lachnospiraceae*. These families were detected in all samples. *Enterobacteriaceae*, *Prevotellaceae*, *Peptostreptococcaceae*, and *Clostridiaceae* were found in only three of the four data sets. These seven bacterial families each accounted for over 60 percent of the microbiota (see Figure 2).

## 3. Discussion

In the present study, the prevalence of ESBL-producing *E. coli* in wild boar and wild ruminants is significantly lower compared to recent studies. However, the studies differed in sample size and methodology, limiting comparability. 

However, this match could only be confirmed for the samples from 2020. In the framework of the German National Zoonoses Monitoring Program, samples from hunted wild boar and roe deer were collected nationwide in 2016 and 2017, respectively. Selective isolation of ESBL-/AmpC-producing *E. coli* yielded 6.5% (36 of 551) positive samples from wild boar (2016) and 2.3% (13 of 573) from roe deer (2017) [35]. Only for the samples from wild ruminants from 2020, we obtained comparable results. Similar findings were reported in a study by Hortmann et al. [21]. They collected fecal samples in 22 different hunting areas in Germany between late 2014 and early 2015 and found 5.1% ESBL-positive *E. coli* samples (19/375) (either *bla _CTX-M-1_* or *bla _SHV-12_*). 

A study conducted in Poland included a total of 660 fecal samples collected during two winter seasons from game animals shot during 46 hunts (2012–2013, *n* = 7 and 2013–2014, *n* = 39). The hunts took place in 37 forests distributed across the territory of Poland (wild boar (*n* = 332), red (*n* = 225), roe (*n* = 76), and fallow (*n* = 24) deer. Of 553 *E. coli* isolated, 11 were cefotaxime resistant. Nine of them were retrieved from wild boar (2.7%, range 1.0–4.5%), fallow deer or red deer (a single isolate each) [36].

A significantly higher estimated prevalence of 15.96% of ESBL/AmpC-producing *E. coli* was determined in a study from southern Europe in a total of 1504 fecal samples from wild boar. This study was performed in northern Italy and consisted of four hunting areas comprising approximately 1.800 km^2^. Samples were collected in three consecutive hunting years between 2018 and 2020 [37].

For the future, it would be useful to compare the estimated prevalences of the individual studies in a systematic review. Such a meta-analysis of ESBL-*E. coli* in wildlife would help to better assess the AMR burden in wildlife. Due to the convergence of habitats, the contact of wildlife with other animals (e.g., livestock) and humans becomes more and more common, which subsequently leads to enhanced exchange of antibiotic resistance both in the context of actual bacterial isolates but also mobile genetic elements. As previously mentioned, it appears that wild animals that live in close proximity to human populations carry higher levels of AMRB and AMRG [35,38]. This could for example happen through surface water or agricultural field contamination [3,39]. A recently published study draws a similar conclusion. The authors postulate that wildlife represents a previously unrecognized medium through which environmental ARMGs can be transferred to human clinical pathogens [40].

Analysis of the whole-genome sequence data revealed broad agreement of phenotypic and genotypic results. For the observed phenotypic resistances, genotypic resistance determinants were always found as well (Table 2). Details of the individual whole-genome sequenced *E. coli* are summarized below.

In our study, we detected an ESBL-genotype for all isolates (CTX-M-1, CTX-M-55) phenotypically resistant against cefotaxime. Even if only ESBL-bearing isolates from the above-mentioned studies are taken into account, the detections in our study are still significantly lower than those of other studies. The sampling areas are located in Mecklenburg-Western Pomerania in the northeast of Germany. This region has the lowest human population density in Germany. The sampling areas thus represent a rather secluded part of nature. The associated low contact intensity between humans and agriculture could be an explanation for the low detection frequencies. Unexpectedly, the majority of ESBL-positive isolates was found in 2020 (16/19) and only a small number in 2021 (3/19). In February 2021, it was very cold (partly −15 °C) and there were heavy snowfalls (average temperature: 0.18 °C). In contrast, February 2020 was rather mild (average temperature: 5.50 °C) [41]. It is possible that the low temperatures in 2021 led to the faster death of bacteria in the feces of the sampled animals, so that cultivation was less successful. 

It is of course hardly possible to draw conclusions about the health status of the animals based solely on the nature of the fecal sample. However, it can at least be stated that no intestinal problem, such as diarrhea, was present at the time of defecation, since the consistency of the fecal sample was recorded. Consequently, the sequence analysis of the isolates revealed that they all belong to commensal phylogroups (A, B1, and E). In a recent study of commensal *E. coli* from human fecal samples, ElBaradei et al. found high abundances of ESBL carriers or even multidrug resistance. This is alarming, since these strains are associated with an increased risk of infection [42]. Similar *E. coli* clones belonging to the commensal phylogroup A were increasingly reported in association with ESBL production [43]. Over the past two decades, ESBL have spread rapidly among commensal and pathogenic *E. coli* strains in humans, domestic animals, and environmental sources, making them a substantial public health threat [44].

Sequence analyses of the 19 ESBL-bearing *E. coli* isolates revealed five different MLST sequence (ST) types (ST216, ST398, ST543, ST2768, and ST6448). ST216 was detected most frequently, with nine of the 19 isolates belonging to this ST. *E. coli* ST216 is known to carry resistance genes to a variety of antibiotics and has been detected several times in nosocomial infections [45,46]. In particular, in the context of carbapenem resistance, this ST has appeared several times both in humans [47] and animals [48]. However, we did not detect phenotypic or genotypic carbapenem resistance. Nevertheless, ST216 *E. coli* seems to be particularly capable of capturing and transmitting genes encoding beta-lactamases and carbapenemases [48]. The detection of this ST in more than half of our ESBL isolates is therefore a matter of concern. The second most common ST was ST398 with 21% (four of 19 isolates) and was detected sporadically in both humans and animals as ESBL carriers [49,50,51]. Unlike Holtmann et al., we did not detect any of the globally successful STs [52]. In their study of wild boar, two of the seven sequenced ESBL-carrying *E. coli* belonged to these STs (ST131 and ST648) [21]. 

In our study, IncN was the most abundant inc/rep type, with 16 of the 19 isolated ESBL-*E. coli* carrying this plasmid. CTX-M-1 is frequently associated with IncN, which occurs throughout Europe and is mainly isolated from *E. coli* from animal sources [32]. In particular, this combination is common in *E. coli* from the fecal microbiota of pigs [44]. Other frequent types in our study were IncF plasmids, either with the FIA (*n* = 9) or FIB (*n* = 11) replicon. FII replicons (*n* = 2) were less common. In general, IncF is the most abundant plasmid and predominantly detected in *E. coli*. The resistance genes most frequently described in combination with IncF plasmids are ESBL genes [32]. In our study, we detected ColRNAI plasmids in two isolates in addition to IncFIB and IncFII. These colocinogenic plasmids occasionally carry ESBL genes, but are mostly associated with quinolone resistance genes [53].

A subsequent microbiome analysis performed using CCMetagen [34] revealed similar compositions for both animal species with a dominance of Gram-negative anaerobic genera (*Bacteroidaceae*, *Oscillospiraceae*, and *Lachnospiraceae*) (Figure 2). Interestingly, representatives of *Tannerellaceae* and *Porphyromonadaceae* were present only in the fecal samples of wild boar. However, a reliable statement on this is hardly possible, since it would be based on the results obtained with only two samples. The few microbiome data available in the literature show similar distributions for wild boar. This supports the suitability of our approach for a microbiome analysis based on metagenome data. To the best of our knowledge, no corresponding data are so far available for wild ruminants, so that a comparison is not yet possible. However, we believe that our approach is also suitable for wild ruminant metagenomes. Regarding the analysis of the resistome from metagenomic data, significantly lower normalized counts for AMRG were present in our fecal samples compared with farm animals [27]. When comparing wild boar and wild ruminants, the AMRG levels were even lower in wild ruminants. Wild boar as a cultivator is probably more exposed [6]. 

Only a few publications are available that allow a comparison of data on the resistome of wild boar. Skarzynska et al. applied a metagenomic approach to investigate the resistome of different wildlife species. They found a distribution of each antibiotic group in wild boar that was largely consistent with the distribution in our study (Skarzynska et al. vs. this study): Aminoglycosides: 12.8% vs. 8.3–10.6%; beta-Lactams 9.0% vs. 5.8–6.9%; Glycopeptides 3.8% vs. 0.7–1.2%; Macrolide 9.0% vs. 17.5–17.9% (part of MLS Class); Nitroimidazole 2.6% vs. not present; Quinolone 1.3% vs. 1.3–1.4%; Sulfonamides 6.4 vs. 0.5%; Tetracyclines 56.4% vs. 28.1–29.2%) ([28], Table 2). Despite the broad agreement between the results of the two studies, there are substantial differences in the study designs. The metagenome data from Skarzynska et al. were derived from a pool of ten fecal samples in total, whereas we generated independent metagenomes from individual fecal samples. 

The study of Skarzynska et al. provides resistome data for other wild animals (foxes and rodents), but wild ruminants were not included. Therefore, our resistome data for wild ruminants, as well as the microbiome data, have to remain without comparison with data from other studies, at least for the present moment. However, the high similarity between the wild pig and wild ruminant resistomes in our study is striking (Table 2). Nevertheless, the wild ruminant resistome data presented here can only give a first impression and need to be verified in the future. 

In addition to the limited comparability of the prevalences found in the individual studies, another limitation of our study is the selection of antibiotics tested in the AST. It is only a selection of the available and used substances. In conclusion, our study revealed a detectable but low prevalence of ESBL-producing *E. coli* and an overall low AMR gene burden in wildlife. Shotgun metagenomics revealed significantly lower normalized AMR genes levels in wildlife samples compared to farm animals. In this regard, this confirms our research hypothesis. However, due to its secluded location, the sampling site is a well-known hunting area and therefore an attractive source for acquiring wild game meat. In addition, the forest provides an excellent recreation area for humans [54]. The interconnectedness of wildlife, humans and livestock attracts growing attention by the international organizations such as the FAO [55]. Especially, the consumption of wild game meat is always a reflection of its habitat and is promoted as healthy and pharmaceutical-free alternative to farmed meat [56]. Thus, the detection of AMRG of public health relevance in natural habitats, even in low concentrations, is of concern. In particular, wild ruminants are of interest, as they largely avoid contact with humans, unlike synanthropic species such as wild boar, and may act as an unnoticed reservoir for resistant bacteria and AMRG. Concerning the substances tested in AST, newer antibiotics should also be included in future studies.

Therefore, our data provide a first but also concerning glimpse into a so-far under-investigated part of our environment, that definitely deserves more attention in studies of AMR spread. 

## 4. Materials and Methods

### 4.1. Sampling Areas

The collections of fecal samples were carried out in four hunting grounds in Mecklenburg-Western Pomerania. During two collection periods in February 2020 and 2021, the hunting grounds were systematically searched for feces from wild boar and wild ruminants (red deer, roe deer, and fallow deer). The searches were performed along transects covering the entire study area. Only fecal samples clearly belonging to the target species were included in the study. In addition, samples had to be fresh, i.e., no more than 24 h old. Age was determined by shape and surface moisture. Samples that did not meet these criteria were not included in the study. During the sample collections in February 2020, a total of 820 fresh fecal samples were collected. Of these, 127 were from wild ruminants (roe deer, red deer, and fallow deer) and 693 from wild boar. During the February 2021 sample collections, a total of 775 fresh fecal samples were collected. Of these, 152 were from wild ruminants and 623 were from wild boar.

### 4.2. Bacterial Isolation

Fecal samples were streaked on CHROMID ESBL chromogenic medium (MAST Diagnostica, Oldesloe, Germany) using swabs and subsequently incubated at 37 °C overnight. Putative antibiotic-resistant *E. coli* colonies were identified based on colony morphology (red-purple colonies) and subcultured until pure cultures were achieved. The pure cultures were used for further verification and characterization.

### 4.3. Antimicrobial Susceptibility Testing (AST)

To determine the minimum inhibitory concentrations (MIC) of important antimicrobial compounds, e.g., colistin, AST was carried out in a 96-well plate microdilution procedure. To this end, we prepared an inoculum of the *E. coli* isolate to be tested, which was adjusted to a McFarland standard of 0.5 in 0.9% NaCl. The inoculum (50 µL) was then diluted in 11.5 mL Mueller-Hinton II broth (Oxoid, Wesel, Germany). Thereafter, 100 µL of this dilution were transferred into Merlin MICRONAUT S 96-well AST plates for livestock (“E1-301-100: Zoonosis Monitoring”) (Merlin, Bornheim-Hersel, Germany). The results were visually assessed after incubation for 18–24 h at 37 °C. The decision on turbidity and microbial growth was made by trained scientists. Growth was considered to have occurred when turbidity appeared at the bottom of the well. Tests were considered valid only, if growth had also been observed in the internal growth control.

The European Committee for Antimicrobial Susceptibility Testing (EUCAST) Breakpoint Tables for the Interpretation of MICs and Zone Diameters (version 11.0, 2021. http://www.eucast.org, last accessed 1 October 2021) and European Commission Implementing Decision (EU) 2020/1729 were used for the evaluation.

### 4.4. Whole-genome Sequencing (WGS) and Analysis

WGS was performed for the overall 19 ESBL/AmpC suspect *E. coli* isolates. Sequences were generated using the Illumina NextSeq 500/Nova Seq 6000 platform (StarSEQ, Mainz, Germany). DNA extraction was performed using the MasterPure™ DNA Puri-fication Kit for Blood, Version II (Lucigen, Middleton, WI, USA). DNA extraction was performed using the MasterPure™ DNA Purification Kit for Blood, Version II (Lucigen, Middleton, WI, USA) and subsequently quantified using QuBit fluorometer (Thermofisher Scientific, Waltham, MA, USA). DNA samples were then shipped to StarSEQ. Library preparation was performed by StarSEQ as published elsewhere [57], followed by sequencing using 2 × 150 bp paired-end reads.

Using BBDuk from BBTools v. 38.89 (http://sourceforge.net/projects/bbmap/, last accessed 1 October 2021), the following steps were performed with raw sequencing reads: adapter-trimming (k-mer-based trimming using 23-mer down to 11-mer at the right end using the provided adapter references; additional trimming by paired-end read overlap), filtering for contaminants (k-mer-based removal of phiX174 sequences), and quality-trimming (trimming on both sides for regions with quality < 3; re-removal of poly-G tails ≥ 10 bp; maximum number of Ns after trimming: 0; minimum average quality after trimming: 18; minimum length: 32 bp, filtering reads with entropy less than 0.5 to remove reads with low complexity). Quality controlling of all reads (trimmed reads and raw reads) was performed using FastQC v. 0.11.9 (http://www.bioinformatics.babraham.ac.uk/projects/fastqc/, last accessed 1 October 2021). Using the shovill v. 1.1.0 assembly pipeline (https://github.com/tseemann/shovill, last accessed 1 October 2021) in combination with SPAdes v. 3.15.0 [58], de novo genome assemblies were performed. The pipeline subsampled trimmed reads to be assembled with a maximum coverage of 100×. In addition to the polishing step of the Shovill pipeline, the assemblies were subjected to another polishing step. In this step, bwa v. 0.7.17 [59] mapped all trimmed reads back to the contigs. The SAM/BAM files obtained were then sorted and duplicates identified using SAMtools v. 1.11 [60]. For the following variant-calling, Pilon v. 1.23 [61]. was used and the polished assemblies were checked for suspicious assembly metrics (e.g., high number of contigs and genome size, high N50/N90, low L50/L90). CheckM v. 1.1.3 [62] was additionally used to estimate genome completeness and freedom from contamination. Assemblies were analyzed for multi-locus sequence type (MLST) determination and antibiotic resistance/virulence gene detection using the tools mlst v. 2.19.0 (https://github.com/tseemann/mlst, last accessed 1 October 2021) and ABRicate v. 1.0.0 (https://github.com/tseemann/abricate, last accessed 1 October 2021), respectively. Third-party databases (e.g., PubMLST [63], VFDB [64], ResFinder [65], PlasmidFinder [33], BacMet [66], ARG-ANNOT [67], Ecoli_VF (https://github.com/phac-nml/ecoli_vf, last accessed 1 October 2021)) were used for the analyses of both tools.

### 4.5. Metagenomic Sequencing and Analysis of Antimicrobial Resistance Genes

Fecal samples from wild boar (*n* = 2) and roe deer (*n* = 2) were subjected to metagenomic sequencing. For DNA extraction the QIAamp Fast DNA Stool Mini Kit (QIAGEN, Hilden, Germany) was used applying a modified protocol developed by Knudsen et al. [68]. The main modifications to the protocol supplied with the kit to facilitate increasing cell lysis are the addition of a bead-beating step in the beginning of the isolation and the increase of the temperature during the lysis step to 95 °C. 

Library construction and sequencing was performed by LGC Genomics (LGC Genomics GmbH, Berlin, Germany). LGC the Allegro Targeted Genotyping Kit (TECAN) and the plexWell™ 384 Library Preparation Kit (seqWell™). Firstly, Allegro Targeted Genotyping Kit was used for (i) fragmentation, (ii) end-Repair and Adaptor Ligation, and (iii) purification and secondly, plexWell™ 384 Library Preparation Kit was used for plate-specific barcoding. Libraries were amplified in an emulsion PCR for 14 cycles using standard Illumina primers and afterwards libraries were size selected by a size selection on an LMP-Agarose gel, removing fragments smaller than 300 bp and those larger than 800 bp. A final library purification step and quality control of libraries were performed via Fragmentanalyzer (Agilent, Santa Clara, CA, USA) and Qubit. Sequencing was done on an Illumina NovaSeq 6000 SP FC- 2 × 250 bp paired-end read length (Illumina, Santa Clara, CA, USA) following the manufactures’ instructions. Metagenome sequencing data supplied by LGC was then analyzed using AMR++ and the MEGARes v2.0 database [69,70] using default settings [69]. AMR++ facilitated the removal of low-quality bases and sequences (Trimmomatic–Ref), detection of host DNA and resistance genes (BWA-ref) and the removal of host DNA (Samtools-ref). The resulting data set was then subjected to the actual resistome analysis (ResistomeAnalyzer-Ref). AMR++ yielded counts of hits for AMR sequences at the antibiotic class level, information on the mechanism of action, and the resistance gene level. 

For normalization of the number of AMR reads, we analyzed the raw data (fastq datasets) using METAXA2 v. 2.2.3 [71] to enumerate the number of reads mapping to bacterial 16S rRNA genesLi et al. [72] described a formula for calculating of the normalized abundance of ARGs: $$Abundance=\overset{n}{\underset{1}{\sum }}\frac{N\left(AMRseq\right)*L\left(reads\right)/L\left(AMRrefseq\right)}{N\left(16Sseq\right)*L\left(reads\right)/L\left(16Srefseq\right)}$$
where *n* represents each individual ARG in the MEGARes database, *N*(*AMRseq*) indicates the number of reads identified in the analysis using the AMR++ pipeline, and *N*(*16Sseq*) is the number of reads found to map to bacterial 16S rRNA gene reads in the analysis using METAXA2. *L*(*reads*) corresponds to the length of the sequencing reads (and cancels out of the equation, since the length is the same for both AMR reads and 16S reads), *L*(*AMRrefseq*) is the length of the AMR reference gene in the MEGARes database, and *L*(*16Srefseq*) corresponds to length of the 16S rRNA gene sequence (according to the Greengenes database, the average length of a 16S rRNA gene sequence is 1432 bp).

The calculation was done for each sample separately.

### 4.6. Microbiome Analysis

The CCMetagen pipeline was used for accurate taxonomic assignments of the quality-controlled read sets. The CCMetagen pipeline links the two tools KMA (version 1.3.0) and CCMetagen (version 1.2.2) [34,73]. In the first step, KMA was used to perform a read mapping to a reference database (ncbi_nt_kma database, available at https://researchdata.edu.au/indexed-reference-databases-kma-ccmetagen/1371207, accessed 15 July 2021) (the following options were selected: ipe, -mem_mode, -and, -apm f, -ef, and -1t1). Second, a quality filtering step was performed in the pipeline through CCMetagen (default settings were used).

## Figures and Tables

**Figure 1 antibiotics-11-00123-f001:**
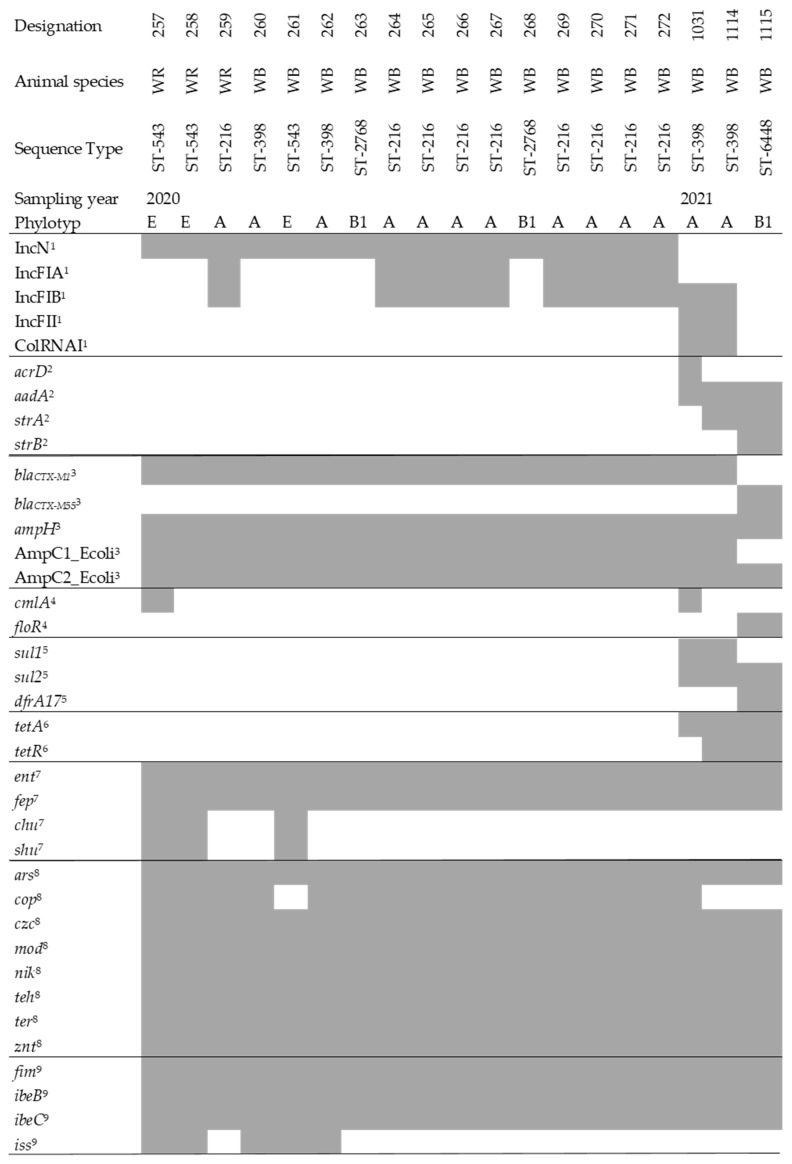
Genotypic characterization of sequenced ESBL-*E. coli* isolates from wild boar (WB) and wild ruminants (WR); presence of a certain factor is based on the results from ABRicate [ABRicate v. 1.0.0 (https://github.com/tseemann/abricate, accessed on 1 October 2021), databases used: VFDB, ResFinder, PlasmidFinder, BacMet, ARG-ANNOT, and Ecoli_VF] using de novo assembled sequences and is depicted in grey. Detected genes are assigned to the following categories: ^1^ Plasmid replicon types, ^2^ aminoglycosides, ^3^ beta-lactam antibiotics, ^4^ phenicol antibiotics, ^5^ sulfonamides and trimethoprim, ^6^ tetracycline antibiotics, ^7^ iron-acquisition related genes, ^8^ heavy metal resistance genes, and ^9^ ExPEC associated genes.

**Figure 2 antibiotics-11-00123-f002:**
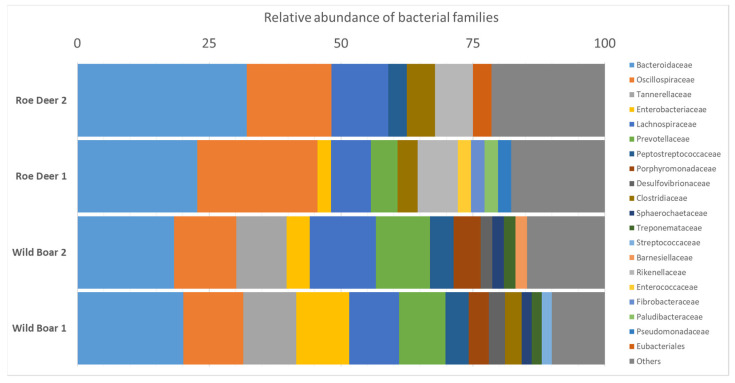
Relative abundance of bacterial families in the microbiome of roe deer and wild boar in percent (%).

**Table 1 antibiotics-11-00123-t001:** Phenotypic resistance profiles of ESBL-*E. coli* isolates obtained from wild boar (WB) and wild ruminants (WR). R = resistant, S = sensitive.

Designation	Species of Origin	Amikacin	Amoxicillin/Clavulanic Acid	Ampicillin	Cefotaxim	Ceftazidim	Cefepim	Colistin	Imipenem	Meropenem	Gentamicin	Tobramycin	Ciprofloxacin	Tetracycline	Trimethoprim/Sulfamethoxazol	ESBL	MDR
257	WR	S	R	R	R	R	R	S	S	S	S	S	S	S	S	+	-
258	WR	S	R	R	R	R	R	S	S	S	S	S	S	R	S	+	+
259	WR	S	R	R	R	R	R	S	S	S	S	S	S	R	S	+	+
260	WB	S	R	R	R	R	R	S	S	S	S	S	S	S	S	+	-
261	WB	S	R	R	R	R	R	S	S	S	S	S	S	R	S	+	+
262	WB	S	R	R	R	R	R	S	S	S	S	S	S	S	S	+	-
263	WB	S	R	R	R	R	R	S	S	S	S	S	S	S	S	+	-
264	WB	S	R	R	R	R	R	S	S	S	S	S	S	S	S	+	-
265	WB	S	R	R	R	R	R	S	S	S	S	S	S	S	S	+	-
266	WB	S	R	R	R	R	R	S	S	S	S	S	S	S	S	+	-
267	WB	S	R	R	R	R	R	S	S	S	S	S	S	S	S	+	-
268	WB	S	R	R	R	R	R	S	S	S	S	S	S	S	S	+	-
269	WB	S	R	R	R	R	R	S	S	S	S	S	S	S	S	+	-
270	WB	S	R	R	R	R	R	S	S	S	S	S	S	S	S	+	-
271	WB	S	R	R	R	R	R	S	S	S	S	S	S	S	S	+	-
272	WB	S	R	R	R	R	R	S	S	S	S	S	S	S	S	+	-
1031	WB	S	R	R	R	R	R	S	S	S	R	R	S	R	S	+	+
1114	WB	S	R	R	R	R	R	S	S	S	R	R	S	R	S	+	+
1115	WB	S	R	R	R	R	R	S	S	S	S	S	R	R	R	+	+

**Table 2 antibiotics-11-00123-t002:** Relative proportions (%) of hits for antibiotic resistance genes summarized on the level of classes of resistance genes according to the analysis of metagenome data with the AMR++ pipeline.

Class	Wild Boar 1	Wild Boar 2	Roe Deer 1	Roe Deer 2
Aminocoumarins	2.0	1.1	3.0	1.5
Aminoglycosides	10.6	8.3	15.2	18.3
Bacitracin	0.7	0.0	0.0	0.0
Beta-lactams	5.8	6.9	8.5	6.6
Cationic antimicrobial peptides	1.4	1.3	0.6	1.2
Elfamycins	1.6	0.8	1.9	2.7
Fluoroquinolones	1.4	1.3	2.6	4.8
Fusidic acid	0.2	0.0	0.6	0.9
Glycopeptides	0.7	1.2	1.1	1.2
MLS	31.1	39.6	27.6	28.8
Multi-drug resistance	11.7	8.7	9.8	9.0
Mycobacterium tuberculosis-specific Drug	0.1	0.3	2.0	1.8
Phenicol	0.4	0.1	0.6	0.9
Rifampin	2.0	1.5	4.5	2.4
Sulfonamides	0.5	0.5	1.7	0.0
Tetracyclines	29.2	28.1	19.3	18.0
Trimethoprim	0.0	0.0	0.2	0.3
Fosfomycin	0.6	0.1	0.9	1.5

**Table 3 antibiotics-11-00123-t003:** Normalized total AMRG abundance.

Wild Boar 1	Wild Boar 2	Roe Deer 1	Roe Deer 2
0.049	0.042	0.033	0.028

## Data Availability

The data for this study have been deposited in the European Nucleotide Archive (ENA) at EMBL-EBI under accession number PRJEB49339 (https://www.ebi.ac.uk/ena/browser/view/PRJEB49339).

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
