# Peer review of "Genomic Analysis of ESBL-Producing E. coli in Wildlife from North-Eastern Germany"

_antibiotics, 2022, doi:10.3390/antibiotics11020123_

Round 1
Reviewer 1 Report
This is a review of the article submitted to the journal Antibiotics entitled “Genomic Analysis of ESBL-producing E. coli in wildlife from north-eastern Germany” by Timo Homeier-Bachmann and colleagues. Overall, this is a very well-thought out study, with a particularly good introduction, that provides adequate context to validate the rationale for the study design as well as to contextualize the main findings in light of similar work.
A few overall suggestions to consider:
- Should MLST typing have been performed on all 1,595 bacterial isolates? Why or why not? MLST typing was performed only on the 19 ESBL isolates, which is clearly important to understand these dangerous isolates, but information on the entire dataset might help with interpreting the relative importance (or evolution?) of the sequence types of the more dangerous ESBL positive isolates. It would be unreasonable to ask for WGS on all 1,595 isolates, but perhaps a simple MLST typing, and/or screening for the presence of IncN and IncF plasmids might be easier?
- The data in Table 2 and Figure 2 is a bit confusing. What is the expected proportion of resistance genes for each antimicrobial class? What is baseline? What is ‘normal’? What is ‘no antibiotic resistance’ and what is ‘the most antibiotic resistance’. If some kind of positive and negative control datasets can be included or referenced here, it would be easier to interpret.
- Table 2 and Figure 2 show the same data in different formats. I don’t think this is appropriate because it is exactly the same data. Perhaps move the Table 2 to Supplemental? The Editor might want to consider how to handle this.
Several edits to both content and language/grammar are recommended as follows:
Line 16: Define ESBL as extended spectrum beta-lactamase
Line 17: Rephrase “as a most critical one” to become “and extended…Enterobacterales are a major contributor”.
Line 25 to 26: This statement comparing AMR rates in the wild animals included in this study with the AMR rates in farm animals does not seem to be justified by the data included and discussed in the paper. In order to make this assertion, it may be necessary to include farm animals from the same region in the actual study. This seems especially true because the results of this study seem to differ from other similar studies in Germany.
Line 38: Replace ‘waters’ with ‘water’
Line 40: Replace the comma with a semicolon before the word ‘however’
Line 58: Replace ‘nowadays’ with ‘currently’
Line 61: Replace ‘are resistant against three’ with ‘are resistant to three’
Line 82: Replace ‘aimed at contributing to filling this’ with ‘aimed to help fill this’
Line 85: Replace ‘isolate-based studies’ with ‘bacterial isolate culture’
Line 87: Replace ‘insight in the’ with ‘insight into the’
Line 99: What is meant by ‘culture-following species such as wild boar’?
Line 105: It seems necessary to mention here again the total number of samples isolated to clarify that 16 (in 2020) and 3 (in 2021) ESBL E. coli isolates identified is not due to a small overall sample size. It is in the abstract, but seems important to repeat in the results section.
Line 107: Replace ‘could be obtained from wild boar’ with ‘were obtained, and these were all isolated from wild boar samples’
Line 108: Replace ‘1.316’ with ‘1,316’
Line 117: Replace ‘form 2021’ with ‘from 2021’
Line 120: Replace ‘definition as multidrug resistant’ with ‘definition of multidrug resistant’
Line 123 and 124 (Table 1 legend): Redo this legend. Remove the reference to colors (there is no grey color). I recommend actually using the label ‘Boar’ and ‘Ruminant’ or ‘Deer’ instead of indicating ruminants with an asterisk.
Line 127: The way this is written suggests that perhaps the authors were doubtful that the samples were indeed E. coli, and therefore had to perform WGS to confirm the identity. If so, then should we also doubt that the non-ESBL isolates were E. coli? This is important to clarify because it suggests that perhaps the 1,595 bacterial samples isolated in this study were a mix of different species.
Line 159: Replace ‘we examined for virulence-associated genes associated with pathogenic bacteria’ with ‘we searched for virulence genes associated with pathogenic bacteria’
Line 167 and 168 (Figure 1 legend): Define the grey versus white color scheme in the figure legend for Figure 1. Does grey indicate presence of the gene? Can you clarify a bit more how ABRicate results are generated?
Line 246-247: Remove “on one side with wildlife on the other side”
Line 261: Replace ‘Alike’ with ‘Similar’
Line 312: Replace ‘6,4’ with ‘6.4’
Lines 351-352: Why was absorbance not quantified with a spectrophotometer? If turbidity was determined with the naked eye, please clarify that and indicate if trained scientists made the decision about turbidity and microbial growth and re-write these sentences accordingly.
Line 430: This LINK no longer exists
Line 446: This accession number does not have associated data in the EMBL-EBI currently.
Reviewer 2 Report
General comments:
The general relevance of the paper is high, understanding the prevalence of antimicrobial resistance genes in natural ecosystems is becoming more important as the AMR bacteria emergency is maintained in public health and livestock.
The authors are profiting from the previous study of Liu et al., which provides the baseline for the present work. The introduction has a good presentation of problems, previous literature, and side goal.
Although complete, I think the introduction should have a wider paragraph explaining the research hypothesis and the main goal, which should be more than “gaining deeper insights”. The authors performed a surveillance of presence of AMR genes in wild populations, and we knew from the literature that the levels were lower in wild populations.
What is the contribution of this work other than a deeper analytical perspective and quantitative data on wild populations? It would be great that the authors address that as an introduction of the paper.
Specific comments:
Introduction:
Lines 51-69: Ok, the relevance of the study is that AMR genes are of high interest for public health and our sanitary emergency. But what are the relationship between prevalence in wild animals and human infections? Is there a contribution to a higher frequency of zoonotic infections? do the fecal samples contaminate human settlements? Is this study important for public health, or just veterinary transmission?
The authors should add a few more lines trying to round up all the presented evidence, with special emphasis on the relationship between livestock-wild animals-public health.
Lines 97-100: The authors take only three sentences to introduce an insight of the research hypothesis and the main goal of the work. Therefore, it becomes harder to test if the conclusions support the hypothesis. It must be more than just getting more data.
Discussion:
Lines 209 – 227: Knowing the prevalence in other areas is important, but I am not sure how relevant is to make comparisons in prevalence and sample sizes to discuss the obtained results.
Lines 240 – 253: So, even considering all the sampling biases, we are still uncertain about the significance of the low frequencies of AMR genes that you found. What is the contribution of this work?
Lines 306 – 324: The study lacks a conclusion, which is not a surprise because it is lacking a research hypothesis, although interesting and relevant, this study does not comply by the editorial rules of MDPI, given the fact that it does not present a conclusion supported by the obtained results.
I see this as a major problem that makes me recommend rejection to the editorial, but with sufficient corrections, the authors may be able to submit again this manuscript to MDPI.
Reviewer 3 Report
The study analysed the ESBL-producing E. coli in German wildlife, intending to report the antimicrobial resistance in the animals. The study is presented well, and the finding is significantly important. A few minor corrections are required in the manuscript.
- The authors have mentioned the treatment of bacterial infections in the abstract and the introduction. The study was conducted on the wild animals' fecal microbiota, which does not affect the animals even if the bacterial strains are resistant. The authors have not established any relationship between the animal microbiota and its dissemination into human beings.
- The fecal specimens were collected from the hunting grounds. How was the environmental contamination avoided?
- The study design, exclusion and inclusion criteria are not mentioned in the methodology section.
- The study's limitations should be included at the end of the discussion; several newer antibiotic molecules were not tested.
- The conclusion must be written with more clarity and should also focus on the antibiotic data, which is the potential aim of the study.
Reviewer 4 Report
In this manuscript, the presence of ESBL-producing E. coli in wild animals, ruminants and wild boars were studied. Genomes, metagenomes, and microbiota were analyzed.
The following comments are made:
- Line 23. What is CTM-X? before putting abbreviations put the full name. Check in all the text.
- Lines 56 and 57. What does TEM and SHV mean?
- Line 60. Who is 3MRGN? What antibiotics are they resistant to?
- Line 108. What does CI mean?
- Line 124. “resistant phenotype is depicted by gray color”. Color not seen in Table 1.
- Figure 1. Could you indicate what the studied genes code for.
- Line 174. What does AMRGs mean?
- Line 211. “from Italy was higher than in ours”. Put reference
- Line 328. Were the environmental conditions in these areas in 2020 and 2021 unaffected by the confinement restrictions of humans due to the COVID-19 pandemic? In other words, human or livestock contamination was not reduced?
- Line 337. Why were other ESBL-producing bacteria not isolated? Why only E. coli?
- Discussion. Discuss how wild animals acquire resistance genes.
- Line 359. "All ESBL / AmpC suspect E. coli were subjected to whole-genome sequencing." How many samples were there?
- Line 390. Why only 2 samples? Why didn't they pool?
- You did not make Conclusions. You must put your conclusions.
Round 2
Reviewer 2 Report
I thank the authors for addressing each one of my concerns, this helped to not only restructure better the manuscript, but to take shape as a relevant investigation.
Reviewer 4 Report
The authors made the suggested changes